# Muscle Radiodensity Reduction in COVID-19 Survivors Is Independent of NLR Levels During Acute Infection Phase

**DOI:** 10.3390/ijerph22040521

**Published:** 2025-03-28

**Authors:** Mônica Aparecida Prata Alves, Fabiana Lascala Juliani, Beatriz Rafaelle Goes-Santos, Maria Carolina Santos Mendes, Mônica Corso Pereira, José Barreto Campello Carvalheira, Lígia M. Antunes-Correa

**Affiliations:** 1Department of Adapted Physical Activity Studies (DEAFA), School of Physical Education, University of Campinas (FEF/UNICAMP), Campinas 13083-851, Brazil; m234748@dac.unicamp.br (M.A.P.A.);; 2Division of Oncology, Department of Anesthesiology, Oncology and Radiology, School of Medical Sciences, University of Campinas (FCM/UNICAMP), Campinas 13083-888, Brazil; 3Department of Internal Medicine, School of Medical Sciences, University of Campinas (FCM/UNICAMP), Campinas 13083-888, Brazil; 4Department of Pulmonology, School of Medical Sciences, University of Campinas (FCM/UNICAMP), Campinas 13083-888, Brazil

**Keywords:** COVID-19 survivors, neutrophil-to-lymphocyte ratio, skeletal muscle radiodensity

## Abstract

COVID-19 survivors often experience late symptoms, possibly secondary to an exacerbated inflammatory response. This study aimed to investigate whether inflammatory levels, assessed by the neutrophil-to-lymphocyte ratio (NLR) during hospitalization in the acute phase of SARS-CoV-2 infection, affect the skeletal muscle phenotype and adipose tissue of COVID-19 survivors during outpatient follow-up after discharge. This retrospective, single-center study included COVID-19 survivors hospitalized from March 2020 to April 2021, who attended outpatient follow-ups 3 to 9 months after discharge. Patients were divided into two groups based on inflammatory levels during hospitalization: (1) low NLR (≤4.2) and (2) high NLR (>4.2). The skeletal muscle phenotype and adipose tissue were assessed using computed tomography. The study included 60 patients: 20 low NLR and 40 high NLR. The high NLR group was unexpectedly younger, but had longer hospital stays and required more intensive care. We observed a reduction in skeletal muscle radiodensity and an increase in skeletal muscle fat in both groups. However, we observed no differences in subcutaneous and visceral adipose tissue between hospitalization and follow-up. We conclude that COVID-19 survivors show reduced skeletal muscle radiodensity and increased skeletal muscle fat infiltration post-hospitalization, regardless of NLR levels during acute infection. In addition, age and intramuscular fat infiltration during hospitalization are associated with reducing skeletal muscle radiodensity. This highlights the need for targeted rehabilitation to address long-term muscle effects and recovery.

## 1. Introduction

Severe acute respiratory syndrome coronavirus 2 (SARS-CoV-2) was swiftly identified as the causative agent of coronavirus disease 2019 (COVID-19), which spread rapidly across the globe, resulting in a worldwide pandemic [1,2]. Beyond its significant impact on incidence, hospitalization, and mortality rates, a substantial proportion of survivors experience prolonged sequelae extending beyond the acute phase of infection. Current evidence suggests that approximately 10% of survivors develop long-term symptoms [3,4]. These symptoms encompass a wide range of clinical manifestations, including dyspnea, fatigue, muscle weakness, cognitive dysfunction, myalgia, pharyngitis, cough, diarrhea, anosmia, and dysgeusia. Additionally, survivors often experience a range of complications, including cardiovascular disorders, thrombotic events, cerebrovascular conditions, type 2 diabetes, dysautonomia, myalgic encephalomyelitis/chronic fatigue syndrome, and skeletal muscle dysfunction [5,6].

In recent years, several researchers have focused on elucidating the pathophysiological mechanisms underlying post-COVID-19 skeletal muscle disturbances. Some studies demonstrate that COVID-19 affects skeletal muscle through various mechanisms, including the direct infection of muscle cells by SARS-CoV-2, as well as indirect muscle damage during both acute infection and recovery phases [7,8]. The inflammatory response and cytokine storm observed during acute SARS-CoV-2 infection significantly contribute to muscle alterations [9]. Pro-inflammatory cytokines disrupt muscle metabolism, promote muscle mass loss, impair satellite cell differentiation, and enhance fibroblast activity [10,11,12]. Additionally, COVID-19 infection induces decreased mobility and physical activity due to prolonged hospitalization, which contributes to muscle loss through disuse atrophy, a well-documented phenomenon [13]. Inadequate nutritional intake during hospitalization also exacerbates muscle degradation, with malnutrition affecting up to 67% of intensive care unit patients [14]. Despite these acute alterations, skeletal muscle can experience long-term disturbances in COVID-19 survivors. Evidence suggests that anomalous microvesicles enriched with inflammatory molecules from the acute phase, which are resistant to fibrinolysis, persist in the blood of individuals with long COVID. The cytokines, which are trapped within microvesicles, are continuously released into circulation, sustaining chronic inflammation and infiltrating various tissues over time [9]. Moreover, SARS-CoV-2 infection induces sustained pro-inflammatory reprogramming, potentially mediated by epigenetic modifications, which may amplify age-related chronic inflammation and exacerbate skeletal muscle alterations during the recovery phase [9].

Several authors also have reported an elevated prevalence of adipose tissue accumulation among patients with COVID-19. Additionally, a phenotype characterized by reduced skeletal muscle area, low muscle radiodensity, and/or excess adiposity has been associated with an increased risk of severe disease and mortality [15,16]. A previous study conducted by our research group demonstrated a significant association between subcutaneous adipose tissue radiodensity and severe COVID-19 outcomes. Patients with increased subcutaneous adipose tissue radiodensity experienced prolonged hospitalizations and a greater need for mechanical ventilation [17]. Elevated adipose tissue radiodensity has been linked to heightened inflammatory activity, such as macrophage infiltration, the activation of other immune system components, and the increased production of adipokines. Collectively, excessive adipose tissue and chronic systemic inflammation contribute to ectopic fat deposition within skeletal muscle [17].

Furthermore, inflammatory biomarkers, particularly the neutrophil-to-lymphocyte ratio (NLR), have been extensively employed to assess systemic inflammation during acute SARS-CoV-2 infection. Our previous research demonstrated that patients with elevated NLR (>4.2) had a significantly higher risk of mortality (OR: 5.00; CI: 1.88–13.55; *p* = 0.001) compared to those with lower NLR (≤4.2) [18]. Given the critical prognostic significance of elevated inflammatory markers, it is plausible that NLR may be associated with long-term skeletal muscle alterations in COVID-19 survivors.

Consequently, the present study aimed to investigate whether inflammatory levels, as assessed by the neutrophil-to-lymphocyte ratio (NLR) during hospitalization in the acute phase of SARS-CoV-2 infection, affect the skeletal muscle phenotype and adipose tissue of COVID-19 survivors during outpatient follow-up after hospital discharge.

## 2. Materials and Methods

### 2.1. Study Design

This retrospective, single-center study involved patients hospitalized for COVID-19 at the Clinical Hospital (HC)—UNICAMP from March 2020 to April 2021. These patients participated in outpatient follow-up appointments 3 to 9 months after hospital discharge, based on specific inclusion and exclusion criteria. The inclusion criteria consisted of patients over 18 years of age with a laboratory diagnosis of SARS-CoV-2 by RT-PCR and a lymphocyte-to-neutrophil ratio assessed at admission or during hospitalization for COVID-19, as well as a computed tomography (CT) scan at the level of L1 during hospitalization for COVID-19 and during outpatient follow-up. Patients who underwent a contrast CT scan, had low image quality, had analysis compromised due to artifacts or ascites, had only one CT scan, or had CT scans conducted outside the 3 to 9 months post-discharge period were excluded, and who lacked important clinical information available (such as date of birth, comorbidities, hospitalization data, etc.) for consultation in their medical records, were excluded from our study. After inclusion, the patients were divided into two groups based on inflammatory levels during hospitalization: (1) low inflammation (NLR ≤ 4.2) and (2) high inflammation (NLR > 4.2). This study was approved by the Human Subject Protection Committee of the University of Campinas, São Paulo, Brazil (CAAE: 31783420.7.0000.5404) and complied with all the guidelines of the Declaration of Helsinki.

### 2.2. Physical, Clinical, and Laboratory Data

Data collection for physical, clinical, and laboratory characteristics was performed through electronic medical records and tabulated in the Research Electronic Data Capture—REDcap platform.

### 2.3. Inflammatory Level: Neutrophil-to-Lymphocyte Ratio

The analysis of inflammatory levels was conducted using the neutrophil-to-lymphocyte ratio (NLR), which was calculated from the absolute serum concentrations of neutrophils (g/dL) and lymphocytes (g/dL). The threshold for group classification was based on the criteria established by Padilha et al., utilizing the Youden index [18], with low inflammation defined as NLR ≤ 4.2 and high inflammation as NLR > 4.2.

### 2.4. Skeletal Muscle Phenotyping and Adipose Tissue Characterization

The muscle phenotype and adipose tissue of COVID-19 survivors were assessed using CT images stored in the HC—UNICAMP Arya medical. Cross-sectional images at the level of the first lumbar vertebra (L1) were analyzed by a single trained and blinded evaluator (F.J.L.) using SliceOMatic V.5.0 software (Tomovision, Montreal, QC, Canada) [19]. The analysis of skeletal muscle included the psoas, quadratus lumborum, paravertebral, latissimus dorsi, intercostal, rectus abdominis, internal and external obliques, and transversus abdominis muscles [20]. The software was configured to analyze skeletal muscle (SM) within a radiodensity range of −29 to +150 Hounsfield units (HU), intramuscular adipose tissue (IMAT) within −150 to −50 HU, visceral adipose tissue (VAT) within −150 to −50 HU, and subcutaneous adipose tissue (SAT) within −190 to −30 HU [20]. Radiodensity attenuation (R, HU), cross-sectional area (A, cm^2^), and tissue index (I, cm^2^/m^2^) were evaluated for all tissue types [21,22].

Additionally, handgrip strength (HGS) was assessed using a dynamometer (JAMAR). Participants were instructed to sit comfortably in an armless chair with their feet flat on the floor and their hips and knees flexed at approximately 90 degrees [23]. The non-dominant hand was placed resting on the ipsilateral thigh. Once properly positioned, participants were instructed to perform three maximal contractions of the dynamometer using their dominant hand [23]. The average of the three recorded values was used for analysis.

### 2.5. Statistical Analysis

The Shapiro–Wilk test was used to assess the normality of the variable distributions. To analyze the characteristics of the groups with low and high NLR during hospitalization, continuous variables were analyzed using an unpaired Student’s *t*-test or an unpaired Mann–Whitney U test and presented as mean and standard deviation or median and interquartile range. Categorical variables were analyzed using chi-squared or Fisher’s exact test and presented as counts and percentages. We employed a two-way ANOVA for repeated measures to analyze potential changes between the hospitalization period and outpatient follow-up. In addition, we conducted univariate and multivariate linear regression using the enter method and skeletal muscle radiodensity during outpatient follow-up as a dependent variable. Statistical analyses were performed using Jamovi 2.6.26 software, with a significance level set at *p* < 0.05.

## 3. Results

We assessed a database of 245 patients hospitalized at HC—UNICAMP from March 2020 to April 2021 who underwent outpatient follow-up after discharge. Of these, 12 patients were excluded because their follow-up CT scans were conducted either earlier than or beyond the 3–9 months post-discharge period. Additionally, 153 patients were excluded due to CT imaging during hospitalization or follow-up being performed with contrast, occurring outside the defined time frame, having low image quality, or being compromised by artifacts or ascites. Consequently, 60 patients underwent CT scans both during hospitalization and at follow-up: 20 with low NLR and 40 with high NLR (Figure 1).

### 3.1. Characteristics of Patients During Hospitalization

The physical, clinical, and laboratory characteristics of the low and high NLR groups during COVID-19 hospitalization are summarized in Table 1. Notably, patients in the low NLR group were older. However, no significant differences were observed in sex distribution, the number of associated comorbidities, or smoking status. The considered comorbidities included cardiac, respiratory, endocrine, rheumatological, digestive, neurological, genitourinary, infectious, psychiatric, and oncological conditions. In addition, when we analyzed the prevalence of comorbidities in isolation, there were no differences in the prevalence of hypertension, diabetes, dyslipidemia, obesity, and other diseases between groups of COVID-19 survivors.

Laboratory analyses revealed that patients in the high NLR group exhibited elevated levels of leukocytes, neutrophils, platelets, and D-dimer. Conversely, as expected, lymphocyte levels were lower in the high NLR group. No significant differences were observed between groups for other laboratory parameters or blood gas analysis results (Table 1).

In terms of treatment characteristics during hospitalization, patients in the high NLR group experienced longer hospital stays, required more advanced ICU care, and had a higher incidence of adverse events. These events included infections, neurological complications (e.g., seizures, stroke, neuropathy), visual disturbances, cardiac issues (e.g., acute myocardial infarction, arrhythmias), thromboembolic or vascular complications (e.g., deep vein thrombosis, peripheral thromboembolism, peripheral ischemia), hematological abnormalities (e.g., leukopenia, anemia, thrombocytopenia, lymphopenia), genitourinary and renal complications (e.g., dialysis-dependent or non-dialysis-dependent renal failure), gastrointestinal complications (e.g., diarrhea, hepatitis), and dermatological issues (e.g., pressure ulcers). No differences were observed between groups in the need for oxygen therapy (Table 1).

### 3.2. Characteristics of Patients During Follow-Up

During the follow-up period (3 to 9 months after hospital discharge), no significant differences were observed between groups in weight, height, body mass index (BMI), handgrip strength, or the length of the follow-up interval (Table 2).

### 3.3. Skeletal Muscle Phenotyping and Adipose Tissue Characterization

Longitudinal analysis of the skeletal muscle phenotype revealed a time-dependent reduction in skeletal muscle radiodensity (SMR) between hospitalization and follow-up, regardless of the inflammatory status (low or high NLR) during hospitalization (Table 3). Conversely, an increase in intramuscular adipose tissue, measured by the intramuscular fat area (IMATA) and intramuscular fat index (IMATI), was observed in both groups, also independent of inflammatory status during hospitalization. However, we did not observe any differences in subcutaneous and visceral adipose tissue between hospitalization and follow-up, and nor were there differences between groups or interactions between time and group status (Table 3).

### 3.4. Linear Regression

Linear regression was additionally performed to see which variables could predict SMR of outpatient follow-up (3 to 9 months post-discharge) in COVID-19 survivors. First we performed a univariate linear regression using variables that achieve significant results (Table 4). As result, we found a negative association between age, sex, IMATA, IMATR, SATA, and SATR with SMR of outpatient follow-up (3 to 9 months post-discharge) in COVID-19 survivors (Table 4). In addition, a positive association was observed between NLR, SMA, and IMATI, with SMR of outpatient follow-up (3 to 9 months post-discharge) in COVID-19 survivors (Table 4).

Given the results of univariate linear regression, we constructed one model using these variables with significant values in multivariate linear regression. The measurement of area and index were correlated, so we constructed two models, with one using area and other significant parameters as predictors (Table 5), and one using index and other significant parameters as predictors of SMR of outpatient follow-up (3 to 9 months post-discharge) in COVID-19 survivors (Table 6).

As the results, we found an inverse association between age and IMATA with SMR of outpatient follow-up (3 to 9 months post-discharge) in COVID-19 survivors (Table 5). In addition, in the second model, we found an inverse association between age and IMATI with SMR of outpatient follow-up (3 to 9 months post-discharge) in COVID-19 survivors (Table 6). Considering this results, it is possible to affirm that higher ages and increase in intramuscular adipose tissue area (IMATA) or index (IMATI) could predict the decrease in skeletal muscle radiodensity of outpatient follow-up (3 to 9 months post-discharge) in COVID-19 survivors (SMR).

## 4. Discussion

In the present study, our findings demonstrate that COVID-19 survivors exhibit a reduction in skeletal muscle radiodensity and an increase in intramuscular fat post-hospital discharge, independent of the inflammatory response at the time of hospital admission for acute-phase SARS-CoV-2 infection. In addition, we observed an association between age and intramuscular fat infiltration during the hospitalization period with the skeletal muscle radiodensity in the follow-up.

Persistent and late-onset muscular alterations in COVID-19 survivors have gained significant attention from the scientific community since the onset of the pandemic. Besutti et al. [24] conducted a retrospective study involving 208 survivors of severe SARS-CoV-2 pneumonia, utilizing CT imaging to assess skeletal muscle parameters at diagnosis, 2–3 months, and 6–7 months post-diagnosis. Their findings demonstrated persistent muscle mass loss over time, especially in individuals with elevated inflammatory markers during the acute phase [24]. In contrast, our study identified reductions in skeletal muscle radiodensity rather than muscle mass loss, with these changes occurring independently of acute-phase NLR levels.

The systemic inflammatory response associated with acute COVID-19, characterized by an exaggerated release of pro-inflammatory cytokines, such as IL-6 and TNFα, may disrupt protein turnover and muscle mass loss. Pro-inflammatory cytokines promote a catabolic state during the acute phase of COVID-19 and exacerbate the imbalance between protein synthesis and degradation, like other conditions that require ICU treatment. However, the exact role of the ubiquitin–proteasome pathway and calpain in COVID-19 remains unclear [25]. Other evidence suggests immune cell involvement in skeletal muscle alterations. A detailed study of skeletal muscle samples from patients who died of severe COVID-19 revealed significant infiltration of leukocytes, T-cells, and natural killer cells compared to critically ill patients without COVID-19, highlighting the role of immune cell involvement in severe cases [26].

The exact mechanisms leading to reduced skeletal muscle radiodensity and increased intramuscular fat infiltration in COVID-19 survivors remain unclear. Emerging evidence suggests that these phenomena may result from both direct and indirect effects of SARS-CoV-2 infection. In addition to systemic inflammation, these effects may encompass muscle tissue damage, related comorbidities, and the impact of clinical management practices [27].

The reduction in SMR may or may not coincide with muscle mass loss, and is partially attributable to increased intramuscular fat infiltration. Intramuscular fat accumulation, typically associated with aging, is linked to increased differentiation of satellite cells into adipocytes, elevated cytokine activity, and enhanced intramuscular lipogenic regulation [28]. Evidence indicates that intramuscular fat disrupts muscle homeostasis, adversely affecting muscle mass regulation, mitochondrial function, metabolic processes, and inflammation [28]. These disruptions impair protein synthesis via the Akt/mTOR pathway, accelerate protein degradation through the ubiquitin–proteasome system, and induce mitochondrial dysfunction, marked by reduced PGC-1α expression and oxidative enzyme activity, impaired lipid oxidation, and heightened production of pro-inflammatory cytokines (e.g., TNFα, MCP-1, IL-6). Collectively, these mechanisms contribute to insulin resistance and glycemic instability, which are significant factors in the accumulation of ectopic adipose tissue in skeletal muscle [27]. The present study demonstrated an association in multivariate linear regression between age and intramuscular fat infiltration during the acute phase of COVID-19 and SMR in outpatient follow-up. Our findings suggest that older patients, as well as those exhibiting alterations in muscle tissue during hospitalization, demonstrate more pronounced late-stage muscle changes.

Mitochondrial dysfunction, a prominent feature of COVID-19, impairs skeletal muscle energy production and disrupts recovery and muscle mass regulation. Girolamo et al. [29] demonstrated that SARS-CoV-2-induced muscular hypoxia exacerbates inflammation and local metabolic disturbances, leading to muscle atrophy [29]. Similarly, Appelman et al. [12] found reduced mitochondrial enzyme activity, lower levels of energy-producing metabolites, and amyloid deposits in muscle biopsies from patients with persistent post-COVID-19 symptoms, further implicating mitochondrial dysfunction and muscle function [12]. Metabolic muscle alterations also may contribute to reduced skeletal muscle radiodensity and increased intramuscular fat infiltration in COVID-19 survivors.

Excess adipose tissue may also play a significant role in these changes. A previous study by our group showed that the radiodensity of subcutaneous adipose tissue is associated with severe COVID-19 outcomes and prolonged hospitalization [17]. It is well established that increased adipose tissue augments the production of inflammatory adipokines through macrophage infiltration and immune activation [17]. Despite the significance of multivariate linear regression, increased subcutaneous adipose tissue could contribute to the reduction in skeletal muscle radiodensity, stimulating the ectopic fat deposition within skeletal muscle and thereby worsening adverse musculoskeletal outcomes.

In addition, hospitalization-related factors, including immobility, stays in ICU, and corticosteroid use may further contribute to alterations in skeletal muscle phenotype. While corticosteroids effectively manage inflammation, their use has been associated with skeletal myopathy [9,30]. Malnutrition may also contribute to significant muscle alterations. Although our study did not directly assess the effects of corticosteroids and malnutrition within our cohort, this remains a crucial area for future research. Finally, a previous study conducted by our group demonstrated a strong association between elevated NLR levels and increased mortality risk in COVID-19 patients [18]. However, contrary to our current hypothesis, this study found that patients with higher NLR levels did not exhibit more pronounced muscle tissue changes. It is crucial to underscore that we focused our analysis on a single inflammatory biomarker, the NLR, owing to its widespread application in clinical practice. Consequently, it remains plausible that our results could have varied had we assessed a different biomarker. Nonetheless, our findings demonstrate that all patients infected with COVID-19 in its most severe form—specifically those requiring hospitalization—exhibited delayed alterations in skeletal muscle. This discovery opens new issues for further research in this area.

### Study Limitations

This study has several limitations. Firstly, being a retrospective, single-center study, it lacked a control group for comparison. Secondly, data collection during the early waves of the pandemic prevented the assessment of vaccination effects, as vaccination rates were low, and vaccination status was unavailable. Thirdly, the absence of nutritional support, anthropometric data (such as weight and height), and functional capacity measurements during hospitalization limited our ability to conduct more comprehensive analyses.

## 5. Conclusions

COVID-19 survivors exhibit decreased skeletal muscle radiodensity and increased skeletal muscle intramuscular fat infiltration following hospital discharge, irrespective of the NLR levels encountered during acute SARS-CoV-2 hospitalization. In addition, age and intramuscular fat infiltration during hospitalization are associated with reducing skeletal muscle radiodensity. These findings underscore the urgent need for specialized rehabilitation strategies to mitigate the long-term effects of COVID-19 on muscle characteristics and overall recovery.

## Figures and Tables

**Figure 1 ijerph-22-00521-f001:**
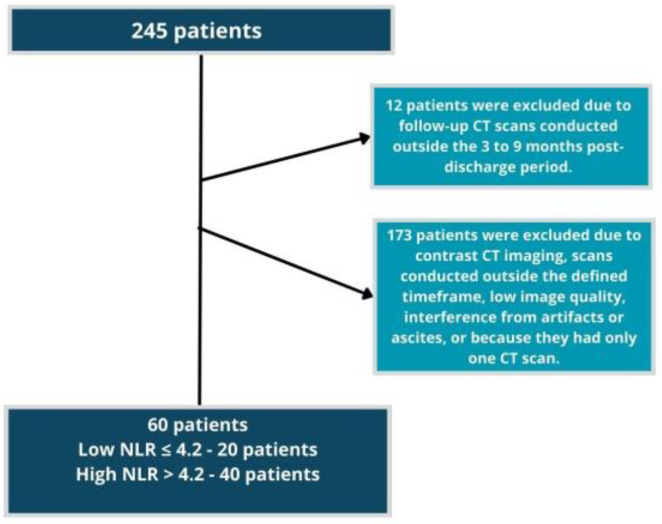
Flowchart showing the recruitment process. CT: computed tomography. NLR: neutrophil-to-lymphocyte ratio.

**Table 1 ijerph-22-00521-t001:** Physical, clinical, and laboratory characteristics of patients according to the neutrophil-to-lymphocyte ratio (NLR) during hospitalization for COVID-19.

	Low NLR *n* = 20	High NLR *n* = 40	*p* Value
**Physical and clinical parameters**			
Age (years)	63.4 ± 11.5	53.4 ± 13.5	**0.006 ^a^**
Male	12 (60)	14 (35)	0.06 ^c^
Comorbidities			
*None*	0 (0)	2 (5)	0.34 ^d^
*1 comorbidity*	8 (40)	19 (47.5)	
*2 comorbidities*	9 (45)	9 (22.5)	
*3 or more comorbidities*	3 (15)	10 (25)	
Hypertension	12 (60)	14 (40)	0.07 ^c^
Diabetes	6 (30)	7 (17.5)	0.27 ^c^
Dyslipidemia	3 (15)	2 (5)	0.19 ^c^
Obesity	7 (35)	17 (42)	0.58 ^c^
Other Diseases	15 (75)	23 (58)	0.18 ^c^
Smoke			
*Never*	10 (50)	20 (52.6)	0.99 ^c^
*Current or former smoker*	6 (30)	11 (28.9)	
*No information*	4 (20)	7 (18.4)	
**Laboratory parameters**			
Hemoglobin (g/dL)	14.1 ± 1.5	14.0 ± 1.6	0.92 ^a^
Hematocrit (%)	42.2 ± 4.2	41.7 ± 4.3	0.72 ^a^
Leukocytes (×10^3^/µL)	5.7 (4.9–8.0)	9.6 (7.4–11.0)	**<0.001 ^b^**
Neutrophils (×10^3^/µL)	3.5 (2.9–5.9)	7.8 (5.4–9.2)	**<0.001 ^b^**
Lymphocytes (×10^3^/µL)	1.6 (1.4–1.9)	0.8 (0.4–0.6)	**<0.001 ^b^**
Platelets (×10^3^/µL)	181 (163–225)	213 (169–287)	0.17 ^b^
CRP (mg/L)	48 (28–79)	87 (31–130)	0.15 ^b^
Troponin (ng/L)	7.4 (4.8–12)	7.3 (5.1–11)	0.94 ^b^
LDH (U/L)	273 ± 60	304 ± 92	0.16 ^b^
D-dimer (ng/mL)	542(450–838)	796 (583–1457)	**0.03 ^b^**
Glucose (mg/dL)	115 (107–167)	134 (115–180)	0.16 ^b^
ALT (U/L)	36 (22–65)	41 (25–51)	0.71 ^b^
AST (U/L)	36 (28–55)	39 (27–53)	0.91 ^b^
Creatinine (mg/dL)	1.0 (0.8–1.1)	0.9 (0.8–1.1)	0.81 ^b^
Urea (mg/dL)	35 (29–43)	36 (27–46)	0.98 ^b^
Lactate (mmol/L)	1.6 (1.2–2.2)	1.7 (1.2–2.1)	0.80 ^b^
PH	7.4 (7.4–7.5)	7.4 (7.4–7.5)	0.29 ^b^
**Arterial blood gas analysis**			
PaO_2_ (mmHg)	64 (55–75)	71 (55–80)	0.29 ^b^
PaCO_2_ (mmHg)	35 (32–39)	33 (30–35)	0.18 ^b^
Bicarbonate (mmol/L)	23 (22–24)	23 (22–24)	0.59 ^b^
**Treatment parameters**			
Total days without mechanical ventilation	4.5 (2.8–8)	7 (5–10)	**0.06**
Number of adverse events	1 (1–2)	2 (1–2)	**0.08**
Hospitalization days	8 (4.8–9.5)	12 (9.0–15)	**<0.001 ^b^**
Oxygen requirement	18 (90)	37 (95)	0.60 ^d^
ICU requirement	2 (10)	19 (47.5)	**0.004 ^c^**

Low NLR ≤ 4.2; High NLR > 4.2. CRP: C-reactive protein; ALT: alanine aminotransferase; AST: aspartate aminotransferase; PaO_2_: partial pressure of oxygen in arterial blood: PaCO_2_: partial pressure of carbon dioxide in arterial blood; ICU: intensive care unit. ^a^ Student’s *t*-test: data presented as mean ± standard deviation; ^b^ Mann–Whitney U test: data presented as median (interquartile range); ^c^ chi-square test: data presented as number (%); ^d^ Fisher’s exact test: data presented as number (%).

**Table 2 ijerph-22-00521-t002:** Physical characteristics of patients according to the neutrophil-to-lymphocyte ratio (NLR) during outpatient follow-up (3 to 9 months post-discharge).

	Low NLR *n* = 20	High NLR *n* = 40	*p* Value
**Anthropometric parameters**			
Weight (kg)	84.5 (76.3–96.3)	81.0 (71.0–97.5)	0.475 ^b^
Height (m)	1.67 (1.6–1.73)	1.70 (1.62–1.75)	0.379 ^b^
BMI (kg/m^2^)	31.0 (27.8–37.3)	29 (26.0–33.0)	0.177 ^b^
Follow-up evaluation time	117 (94–143)	108 (92–145)	0.60 ^b^
**Functional capacity**			
Grip strength	26 (23–37)	32 (25–39)	0.55 ^b^

Low NLR ≤ 4.2; High NLR > 4.2. BMI: body mass index. ^b^ Mann–Whitney U test: data presented as median (interquartile range).

**Table 3 ijerph-22-00521-t003:** Skeletal muscle and adipose tissue assessed by computed tomography in relation to neutrophil-to-lymphocyte ratio (NLR) during hospitalization for COVID-19 and outpatient follow-up (3 to 9 months post-discharge) in COVID-19 survivors.

	Hospitalization		Follow-Up		*p* Value		
	Low NLR *n* = 20	High NLR *n* = 40	Low NLR *n* = 20	High NLR *n* = 40	Time	Group	Interaction
**Skeletal Muscle**							
SMA (cm^2^)	126 ± 29	127 ± 33	126 ± 32	132 ± 30	0.111	0.685	0.218
SMI (cm^2^/m^2^)	47 ± 9.9	45.0 ± 10.6	47 ± 9.9	46 ± 9.0	0.231	0.491	0.162
SMR (HU)	37 ± 8.4	39 ± 9.8	33 ± 8.3	36 ± 8.3	**0.011**	0.210	0.593
**Intramuscular Adipose Tissue**							
IMATA (cm^2^)	11 ± 6.5	10.5 ± 6.3	15 ± 8.8	13 ± 9.2	**<0.001**	0.704	0.594
IMATI (cm^2^/m^2^)	4.3 ± 2.6	3.8 ± 2.3	5.7 ± 3.2	4.8 ± 3.6	**<0.001**	0.354	0.523
IMATR (HU)	−62 ± 6.5	−61 ± 6.5	−64 ± 8.5	−61 ± 5.5	0.363	0.182	0.241
**Visceral Adipose Tissue**							
VATA (cm^2^)	154 ± 81	165 ± 67	155 ± 70	174 ± 78	0.318	0.439	0.341
VATI (cm^2^/m^2^)	55 ± 30	59 ± 22	56 ± 26	62 ± 25	0.283	0.503	0.542
VATR (HU)	−96 ± 3.1	−94 ± 9.1	−97 ± 5.7	−95.4 ± 6.6	0.285	0.333	0.774
**Subcutaneous Adipose Tissue**							
SATA (cm^2^)	164 ± 73	177 ± 118	168 ± 74	187 ± 122	0.079	0.537	0.704
SATI (cm^2^/m^2^)	63 ± 35	65 ± 43	67 ± 36	68.6 ± 45	0.118	0.992	0.428
SATR (HU)	−94 ± 8.8	−93 ± 11.1	−98 ± 7.9	−90 ± 29	0.694	0.439	0.490

Low NLR ≤ 4.2; High NLR > 4.2, SMA: skeletal muscle area, SMI: skeletal muscle index, SMR: skeletal muscle radiodensity, IMATA: intramuscular adipose tissue area, IMATI: intramuscular adipose tissue index, IMATR: intramuscular adipose tissue radiodensity, VATA: visceral adipose tissue area, VATI: visceral adipose tissue index, VATR: visceral adipose tissue density, SATA: subcutaneous adipose tissue area, SATI: subcutaneous adipose tissue index, SATR: subcutaneous adipose tissue density. Data presented as mean ± standard deviation for two-way ANOVA.

**Table 4 ijerph-22-00521-t004:** Univariate linear regression using skeletal muscle radiodensity (SMR) of outpatient follow-up (3 to 9 months post-discharge) in COVID-19 survivors as the dependent variable.

Variables	β	95% IC	Adjusted r^2^	*p*-Value
**Physical and Clinical Parameters**				
Age, y	−0.379	−0.381–−0.083	0.129	0.003
Sex (m/f)	−4.308	−12.046–−4.403	0.229	<0.001
NLR, units	0.311	0.041–0.372	0.081	0.015
Diseases	-	-	-	0.074
Smoking	-	-	-	0.365
Drinking	-	-	-	0.138
**Skeletal Muscle**				
SMA (cm^2^)	0.306	0.015–0.148	0.078	0.017
SMI (cm^2^/m^2^)	-	-	-	0.138
**Intramuscular Adipose Tissue**				
IMATA (cm^2^)	−0.476	−0.925–−0.320	0.213	<0.001
IMATI (cm^2^/m^2^)	0.277	0.032–0.683	0.061	0.032
IMATR (HU)	−0.514	−2.612–−0.942	0.249	<0.001
**Visceral Adipose Tissue**				
VATA (cm^2^)	-	-	-	0.478
VATI (cm^2^/m^2^)	-	-	-	0.350
VATR (HU)	-	-	-	0.449
**Subcutaneous Adipose Tissue**				
SATA (cm^2^)	−0.321	−0.046–−0.006	0.087	0.013
SATI (cm^2^/m^2^)	-	-	-	0.064
SATR (HU)	−0.386	−0.132–−0.026	0.132	0.004
**Treatment Outcomes**				
Total days without mechanical ventilation	-	-	-	0.214
Number of adverse events	-	-	-	0.132
Hospitalization days	-	-	-	0.094

IC: confidence interval, NLR: neutrophil-to-lymphocyte ratio, SMA: skeletal muscle area, SMI: skeletal muscle index, SMR: skeletal muscle radiodensity, IMATA: intramuscular adipose tissue area, IMATI: intramuscular adipose tissue index, IMATR: intramuscular adipose tissue radiodensity, VATA: visceral adipose tissue area, VATI: visceral adipose tissue index, VATR: visceral adipose tissue density, SATA: subcutaneous adipose tissue area, SATI: subcutaneous adipose tissue index, SATR: subcutaneous adipose tissue density. Univariate Linear Regression using enter method and skeletal muscle radiodensity as the dependent variable.

**Table 5 ijerph-22-00521-t005:** Multivariate linear regression using skeletal muscle radiodensity (SMR) of outpatient follow-up (3 to 9 months post-discharge) in COVID-19 survivors as the dependent variable (based on area and other significant parameters).

Variables	β	95% IC	*p*-Value
**Age, years**	**−0.328**	**−0.343–−0.058**	**0.007**
Sex, (m/f)	−0.208	−10.167–3.220	0.303
NLR, units	0.067	−0.100–0.l89	0.540
SMA (cm^2^)	0.093	−0.069–0.119	0.600
**IMATA (cm^2^)**	**−0.493**	**−1.121–−0.169**	**0.009**
IMATR (HU)	−0.262	−0.747–0.071	0.103
SATA (cm^2^)	−0.122	−0.039 −0.020	0.507

NLR: neutrophil-to-lymphocyte ratio, SMA: skeletal muscle area, SMR: skeletal muscle radiodensity, IMATA: intramuscular adipose tissue area, IMATR: intramuscular adipose tissue radiodensity, SATA: subcutaneous adipose tissue area. Multivariate linear regression using enter method and skeletal muscle radiodensity as the dependent variable. Model adjusted r^2^ from regression: 0.40.

**Table 6 ijerph-22-00521-t006:** Multivariate linear regression using skeletal muscle radiodensity of outpatient follow-up (3 to 9 months post-discharge) in COVID-19 survivors as the dependent variable (based on index and other significant parameters).

Variables	β	95% IC	*p*-Value
**Age, y**	**−0.310**	**−0.338–−0.038**	**0.015**
Sex (m/f)	−0.222	−8.327–0.995	0.120
NLR, units	0.048	−0.118–0.178	0.681
**IMATI (cm^2^/m^2^)**	**−0.568**	**−3.478–−0.452**	**0.012**
IMATR (HU)	−0.345	−0.874–0.013	0.057
SATI (cm^2^/m^2^)	−0.121	−0.097–0.047	0.493

NLR: neutrophil-to-lymphocyte ratio, IMATI: intramuscular adipose tissue index, IMATR: intramuscular adipose tissue radiodensity, VATA: visceral adipose tissue area, VATI: visceral adipose tissue index, SATI: subcutaneous adipose tissue index, SATR: subcutaneous adipose tissue density. Univariate linear regression using enter method and muscle radiodensity as the dependent variable. Multivariate linear regression using enter method and skeletal muscle radiodensity as the dependent variable. Model adjusted r^2^ from regression: 0.40.

## Data Availability

The data sets presented in this article are not readily available for ethical reasons. Requests for access to datasets should be directed to ligiamac@unicamp.br.

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
