# Peer review of "Muscle Radiodensity Reduction in COVID-19 Survivors Is Independent of NLR Levels During Acute Infection Phase"

_ijerph, 2025, doi:10.3390/ijerph22040521_

Round 1
Reviewer 1 Report
Comments and Suggestions for Authors
The article is devoted to a revelant topic of medicine. Article describes changes in skeletal muscles observed during COVID-19 infection and hypothethasizes about their relevance for prognosis of disease course. Article has a clear goal given at the end of the "Introduction" section.
Materials and methods are described in sufficient detail with exception of statistical methods. Correlation test should be mentioned in "Materials and methods" section and not in results section.
Please move lactate and bicarbonate parameters given in table 1 from "Arterial blood gas analysis" to other section.
Authors should widen the discussion section in regards of possible influence of pro-inflammatory cytokines on skeletal muscle biology during COVID-19. ALso, in results authors showed that NLR and SMD correlate with r = 0,311 with significant p level (p=0,015). Authors should throughly discuss this phenomenon before making statement in conclusion "COVID-19 survivors exhibit decreased skeletal muscle radiodensity and increased skeletal muscle intramuscular fat infiltration following hospital discharge, irrespective ? of the inflammatory levels encountered during acute SARS-CoV-2 hospitalization". Because the absence of relationship between NLR and SMD may depend on the state of immune system of the patient (presence of immunosuppressive disorders, presence of arterial hypertension).
Author Response
Comments 1: The article is devoted to a relevant topic of medicine. Article describes changes in skeletal muscles observed during COVID-19 infection and hypothethasizes about their relevance for prognosis of disease course. Article has a clear goal given at the end of the "Introduction" section.
Response 1: Thank you for the positive comments.
Comments 2: Materials and methods are described in sufficient detail with exception of statistical methods. Correlation test should be mentioned in "Materials and methods" section and not in results section.
Response 2: Thank you for your comment. Following the suggestions from reviewers two and three, we have excluded the correlation and included univariate and multivariate linear regression.
Comments 3: Please move lactate and bicarbonate parameters given in table 1 from "Arterial blood gas analysis" to other section.
Response 3: Thank you for your comment. We have moved lactate and bicarbonate from the 'Arterial Blood Gas Analysis' section to the 'Laboratory Parameters' in Table 1.
Comments 4: Authors should widen the discussion section in regards of possible influence of pro-inflammatory cytokines on skeletal muscle biology during COVID-19. ALso, in results authors showed that NLR and SMD correlate with r = 0,311 with significant p level (p=0,015). Authors should throughly discuss this phenomenon before making statement in conclusion "COVID-19 survivors exhibit decreased skeletal muscle radiodensity and increased skeletal muscle intramuscular fat infiltration following hospital discharge, irrespective? of the inflammatory levels encountered during acute SARS-CoV-2 hospitalization". Because the absence of relationship between NLR and SMD may depend on the state of immune system of the patient (presence of immunosuppressive disorders, presence of arterial hypertension).
Response 4: Thank you for your comment. Following the reviewers' suggestions, we excluded the correlation and included univariate and multivariate linear regression to clarify the results. Tables 04, 05, and 06 present the linear regression results. In addition, we have revised the Discussion and Conclusion, and all changes are highlighted in red for your review - Ln 268 and 359.

Reviewer 2 Report
Comments and Suggestions for Authors
Alves MAP et al report on the influence of COVID-19 survivors on muscle structure, with a particular on inflammatory biomarkers. The study has a decent size (n=80 patients) from which some trends could be observed. The introduction is fine, if a little long. The study is a necessary replication of previous work by other groups.
Major issues.
The analysis was made by stratifying patients by Neutrophil to Lymphocyte Ratio (NLR) - high and low. When this is done however, there is a significant difference in ages (p=0.006), and strong trend for sex (p=0.06).
>The authors need to enlist the help of a statistician to perform a multivariate analysis. This is something that could be done by a post graduate student biostatistician. Given that n=80 (or 60 in some places) it is possible that the data may yield significant findings.
Another option would be compare change in values for each group and compare the difference.
Minor issues
Ln 32 The sentence "We observed..." can have the numbers removed and replaced with a simple statement of findings.
Ln 127
The threshold for NLR was not particurlarly well justified in Padilha et al (which contains some overlap with the current author list). Why is this number used? Why is there no buffer between the two numbers to exclude patients that are very close to the cutoff?
Ln 166
Figure 1 says 245 patients were in the original cohort but 12+173 were not suitable, but this leaves 60 patients and not 80 as indicated. Was there an overlap between the 12 and 173 excluded?
Tables/Data
The authors have run many tests but it appears they have let each comparison stand alone instead of correcting for repeated tests. It may be helpful to mention somewhere that the authors considered each test to be independent.
Table 3 page 9
Some numbers have a full stop/period for a decimal point whereas others have a comma.
Table 4 Ln 231.
Correlations can be strongly influenced by outlying data. The authors should plot these data or better inform reader regarding the distribution.
Author Response
General Comments
Comments 1: Alves MAP et al report on the influence of COVID-19 survivors on muscle structure, with a particular on inflammatory biomarkers. The study has a decent size (n=80 patients) from which some trends could be observed. The introduction is fine, if it’s a little long. The study is a necessary replication of previous work by other groups.
Response 1: Thank you for the comments.
Major issues
Comments 1: The analysis was made by stratifying patients by Neutrophil to Lymphocyte Ratio (NLR) - high and low. When this is done however, there is a significant difference in ages (p=0.006), and strong trend for sex (p=0.06). The authors need to enlist the help of a statistician to perform a multivariate analysis. This is something that could be done by a post graduate student biostatistician. Given that n=80 (or 60 in some places) it is possible that the data may yield significant findings. Another option would be compare change in values for each group and compare the difference.
Response 2: Thank you for your comment. Following the reviewers' suggestions, we excluded the correlation and included univariate and multivariate linear regression to clarify the results. Tables 04, 05, and 06 present the linear regression results. In addition, we have revised the Discussion and Conclusion, and all changes are highlighted in red for your review - Ln 268 and 359.
Minor issues
Comments 2: Ln 32. The sentence "We observed..." can have the numbers removed and replaced with a simple statement of findings.
Response 2: We agree and rewrite the sentence Ln 32: “We observed a reduction in skeletal muscle radiodensity and an increase in skeletal muscle fat in both groups.”
Comments 3: Ln 127. The threshold for NLR was not particularly well justified in Padilha et al (which contains some overlap with the current author list). Why is this number used? Why is there no buffer between the two numbers to exclude patients that are very close to the cutoff?
Response 3: Thank you for your comment. We want to clarify our decision to use this specific cutoff point. The cutoff in the Padilha et al. study was calculated using the Youden index, as referenced in “Fluss, R., Faraggi, D. & Reiser, B. Estimation of the Youden index and its associated cutoff point. Biom. J. 47(4), 458–472 (2005).” This method allows for determining the optimal value for a specific marker within a cohort. Padilha et al. demonstrated that patients with a high NLR > 4.2 were associated with an increased risk of mortality. Our study investigated a similar cohort to assess whether this analogous cutoff point could indicate changes in the body composition of COVID-19 survivors. It is essential to mention that in our cohort, only two patients had a ratio of exactly 4.2. Thus, we included them in our sample and used a cutoff of NLR ≤4.2. We believe that two subjects do not significantly impact the findings of our study. We clarify this information on Ln 126: “The threshold for group classification was based on the criteria established by Padilha et al., utilizing the Youden index.”
Comments 4: Ln 166. Figure 1 says 245 patients were in the original cohort but 12+173 were not suitable, but this leaves 60 patients and not 80 as indicated. Was there an overlap between the 12 and 173 excluded?
Response 4: Thank you for your detailed review. We correct figure 1.
Comments 5: Tables/Data. The authors have run many tests but it appears they have let each comparison stand alone instead of correcting for repeated tests. It may be helpful to mention somewhere that the authors considered each test to be independent.
Response 5: Thank you for your comment. We have rewritten the “Statistical Analysis” section to clarify the tests we have used. Ln 148: “…To analyze the characteristics of the groups with low and high NLR during hospitalization, continuous variables were analyzed using unpaired Student's T-test or unpaired Mann-Whitney U test and presented as mean and standard deviation or median and interquartile range. Categorical variables were analyzed using the Chi-squared test or Fisher’s exact test and presented as counts and percentages. To analyze potential changes between the hospitalization period and outpatient follow-up, we employed a two-way ANOVA for repeated measures…”
Comments 6: Table 3 page 9. Some numbers have a full stop/period for a decimal point whereas others have a comma.
Response 6: Thank you for your detailed review. We correct Table 3 Ln216.
Comments 7: Table 4 Ln 231. Correlations can be strongly influenced by outlying data. The authors should plot these data or better inform reader regarding the distribution.
Response 7: Thank you for your comment. As previously mentioned, following the reviewers' suggestions, we excluded the correlation and included univariate and multivariate linear regression to clarify the results. Tables 04, 05, and 06 present the linear regression results. In addition, we have revised the Discussion and Conclusion, and all changes are highlighted in red for your review - Ln 268 and 359.

Reviewer 3 Report
Comments and Suggestions for Authors
Muscle Radiodensity Reduction in COVID-19 Survivors is Independent of Inflammatory Levels During Acute Infection Phase
Nice work please address below comments:
Abstract:
Please denote what is HU in [low NLR: 37±8.4 vs. 33±8.3 (HU), high NLR: 39±9.8 vs. 36±8.3 (HU), p=0.011].
What are those group comparison referred to in the low NLR population as in 37±8.4 vs. 33±8.3 or in the high NLR population – same comment through the abstract. I thought comparison should be between low and high NLR, but I found two values in each group -clarify more please.
Introduction:
Line 78: Several authors also have reported an elevated prevalence of adipose tissue accumulation among patients with COVID-19 – add references please.
Materials and methods:
Patients who underwent a CT scan with contrast, outside the defined cut-off, with low image quality, or whose analysis was compromised due to artifacts or ascites, and who lacked important clinical information available for consultation in their medical records, were excluded from our study – What is the defined cut off? What is important clinical information included?
Have you considered co-morbidities such as hypertension, DM, obesity, kidney or cardiac conditions or others? I meant to be excluded as it may skew results.
What is the cut off level for low and high NLR based on?
Figure 1: please double check – patient number included and mentioned in the text is total 60 divided to 20 NLR and 40 NLR in the fig. it states 80 divided to 20 and 60!
Results:
Table 1: as age show significant differences – would be helpful to carry a correlation or regression between age and NLR levels.
Could you include BMI values at the baseline stage – it might have effect on NLR and skeletal or adipose tissue phenotype.
I think co-morbidities need to be stratified based on the affected organ – for e.g, heart disease or DM is known to affect skeletal muscle and inflm. markers.
Table 4 show significant correlation between SMD and Age – to exclude that it is the confounding factor affecting the result we may need to include more patient to have them equally distributed between both groups or at it has not significant correlation with NLR and inflammation levels.
Thanks
Author Response
Abstract:
Comments 1: Please denote what is HU in [low NLR: 37±8.4 vs. 33±8.3 (HU), high NLR: 39±9.8 vs. 36±8.3 (HU), p=0.011]. What are those group comparison referred to in the low NLR population as in 37±8.4 vs. 33±8.3 or in the high NLR population – same comment through the abstract. I thought comparison should be between low and high NLR, but I found two values in each group -clarify more please.
Response 1: Thank you for the comment. To clarify the sentence and in accordance with review two, we rewrite the sentence Ln 32: “We observed a reduction in skeletal muscle radiodensity and an increase in skeletal muscle fat in both groups.”
Introduction
Comments 2: Introduction: Line 78: Several authors also have reported an elevated prevalence of adipose tissue accumulation among patients with COVID-19 – add references please.
Response 2: Thank you for the suggestion. We excluded the reference: “Raynard, B. et al., 2022” and included the reference “Hinojosa-Gutiérrez LR et al., 2025. Ln 78 and Ln 371.
Materials and methods:
Comments 3: Patients who underwent a CT scan with contrast, outside the defined cut-off, with low image quality, or whose analysis was compromised due to artifacts or ascites, and who lacked important clinical information available for consultation in their medical records, were excluded from our study – What is the defined cut off? What is important clinical information included?
Response 3: Thank you for your comment. We rewrite the exclusion criteria to clarify this information. Ln 108: “Patients who underwent a contrast CT scan, had low image quality, had analysis compromised due to artifacts or ascites, had only one CT scan, or had CT scans conducted outside the 3 to 9 months post-discharge period were excluded, and who lacked important clinical information available (such as date of birth, comorbidities, hospitalization data, etc.)”
Comments 4: Have you considered co-morbidities such as hypertension, DM, obesity, kidney or cardiac conditions or others? I meant to be excluded as it may skew results.
Response 4: Thank you for your comment. We appreciate your concern; however, we want to clarify that we did not exclude patients with co-morbidities from our study. We know that co-morbidities are a significant risk factor for severe COVID-19. In fact, within our cohort, only two COVID-19 survivors lack co-morbidities. Excluding patients with co-morbidities would have made conducting this study unfeasible.
Comments 5: What is the cut off level for low and high NLR based on?
Response 5: Thank you for your comment. We want to clarify our decision to use this specific cutoff point. The cutoff in the Padilha et al. study was calculated using the Youden index, as referenced in “Fluss, R., Faraggi, D. & Reiser, B. Estimation of the Youden index and its associated cutoff point. Biom. J. 47(4), 458–472 (2005).” This method allows for determining the optimal value for a specific marker within a cohort. Padilha et al. demonstrated that patients with a high NLR > 4.2 were associated with an increased risk of mortality. Our study investigated a similar cohort to assess whether this analogous cutoff point could indicate changes in the body composition of COVID-19 survivors. We clarify this information on Ln 126: “The threshold for group classification was based on the criteria established by Padilha et al., utilizing the Youden index.”
Comments 6: Ln 166 Figure 1 says 245 patients were in the original cohort but 12+173 were not suitable, but this leaves 60 patients and not 80 as indicated. Was there an overlap between the 12 and 173 excluded?
Response 6: Thank you for your detailed review. We correct figure 1.
Results:
Comments 7: Table 1: as age show significant differences – would be helpful to carry a correlation or regression between age and NLR levels.
Response 7: Thank you for your comment. Following the reviewers' suggestions, we excluded the correlation and included univariate and multivariate linear regression to clarify the results. Tables 04, 05, and 06 present the linear regression results. In addition, we have revised the Discussion and Conclusion, and all changes are highlighted in red for your review - Ln 268 and 359.
Comments 8: Could you include BMI values at the baseline stage – it might have effect on NLR and skeletal or adipose tissue phenotype.
Response 8: Thank you for your suggestion. Due to the clinical protocols established at the Clinical Hospital of UNICAMP during the COVID-19 pandemic, patients were not weighed upon admission. As a result, we do not have BMI data for the hospitalization period; we only have it for the follow-up period. Consequently, we cannot incorporate your suggestion into our manuscript. This limitation is acknowledged in our study, and we have included detailed information to clarify this point. Ln 312: “Thirdly, the absence of nutritional support, anthropometric data (such as weight and height), and functional capacity measurements during hospitalization limited our ability to conduct more comprehensive analyses.”
Comments 9: I think co-morbidities need to be stratified based on the affected organ – for e.g, heart disease or DM is known to affect skeletal muscle and inflm. markers.
Response 9: Thank you for your suggestion. According to the information available in the patient's records, we categorized the comorbidities as follows: hypertension, type 2 diabetes mellitus, dyslipidemia, obesity, and other conditions (such as cardiac, respiratory, endocrine, rheumatological, digestive, neurological, genitourinary, infectious, psychiatric, and oncological conditions). Interestingly, we did not observe differences between the groups of patients with low and high inflammation who survived COVID-19. It is well known that these conditions are risk factors for severe clinical manifestations of the disease; therefore, we understand that we did not observe differences between the groups because we only evaluated patients who survived the hospitalization period. We include this information in Table 1 and describe the results in Ln 175.
Comments 10: Table 4 show significant correlation between SMD and Age – to exclude that it is the confounding factor affecting the result we may need to include more patient to have them equally distributed between both groups or at it has not significant correlation with NLR and inflammation levels.
Response 10: Thank you for your comment. Following the reviewers' suggestions, we excluded the correlation and included univariate and multivariate linear regression to clarify the results. Tables 04, 05, and 06 present the linear regression results. In addition, we have revised the Discussion and Conclusion, and all changes are highlighted in red for your review - Ln 268 and 359.

Round 2
Reviewer 2 Report
Comments and Suggestions for Authors
The authors have adequately addressed the queries outlined in the original review.
Minor proofing issues -
Ln 152 "Student's T-test" Student's t-test
SMD is perhaps not the best abbreviation for Skeletal Muscle Radiodensity, but the authors have been consistent with its use.
Author Response
|
Response to Reviewer 2 – Round 2 |
||
|
Summary |
|
|
|
Thank you very much for taking the time to review this manuscript. Please find the detailed responses below and the corresponding corrections in red in the re-submitted files.
|
||
|
Questions for General Evaluation |
Reviewer’s Evaluation |
Response and Revisions
|
|
Does the introduction provide sufficient background and include all relevant references?
|
Yes |
Thank you for the review |
|
Are all the cited references relevant to the research?
|
Yes
|
Thank you for the review |
|
Is the research design appropriate?
|
Yes
|
Thank you for the review |
|
Are the methods adequately described?
|
Yes
|
Thank you for the review |
|
Are the results clearly presented? |
Can be improved
|
Thank you for the review and comments
|
|
Are the conclusions supported by the results? |
Yes
|
Thank you for the review
|
|
Point-by-point response to Comments and Suggestions for Authors |
||
|
Comments 1: The authors have adequately addressed the queries outlined in the original review. Response 1: Thank you for the review and round 2.
Comments 2: Ln 152 "Student's T-test" - Student’s t-test Response 2: Thank you for the suggestion. We corrected Ln 152 in the manuscript. |
||
|
|
||
|
Comments 3: SMD is perhaps not the best abbreviation for Skeletal Muscle Radiodensity, but the authors have been consistent with its use. Response 3: We appreciate your comments and have accordingly updated the abbreviation throughout the manuscript. The changes are highlighted in red. |
||
